# Indoor Inactivation of SARS-CoV-2 Virus by Liquid Hyperoxygen

**DOI:** 10.3390/pathogens13030244

**Published:** 2024-03-11

**Authors:** Giovanni Barco, Zumama Khalid, Alessandra Pulliero, Claudio Angelinetta, Ubaldo Prati, Alberto Izzotti

**Affiliations:** 1Istituto Internazionale Barco SpA, 56121 Pisa, Italy; giovanni.barco@libero.it (G.B.); ubaldo.prati@libero.it (U.P.); 2Department of Health Sciences, University of Genoa, 16132 Genoa, Italy; zumama.khalid@gmail.com (Z.K.); alessandra.pulliero@unige.it (A.P.); 3Biobasic Europe Laboratory, 20146 Milano, Italy; labtox@biobasiceurope.it; 4Department of Experimental Medicine, University of Genoa, 16132 Genoa, Italy; 5IRCCS Ospedale Policlinico San Martino, 16132 Genoa, Italy

**Keywords:** SARS-CoV-2 virus, COVID-19 infection, oxidative cure, environmental disinfection, prevention

## Abstract

The possible future emergence of new SARS-CoV-2 virus variants pushes the development of new chemoprophylaxis protocols complementary to the unspecific and specific immune-prophylaxis measures currently used. The SARS-CoV-2 virus is particularly sensitive to oxidation, due to the relevant positive electrical charge of its spike protein used as a ligand for target cells. The present study evaluated the safety and efficacy of a new oxidant preparation, liquid hyperoxygen (IOL), to neutralize the SARS-CoV-2 virus. IOL was incubated with throat swabs containing a human-type virus. The samples were then incubated with cells expressing the ACE2 receptor and, therefore, very sensitive to SARS-CoV-2 infection. The ability to neutralize SARS-CoV-2 was determined by assessing the amount of viral nucleic acid inside cells by PCR. The results obtained indicate that IOL, even at considerable dilutions, is capable, after incubation times of less than 30 min and even equal to 5 min, of completely inhibiting SARS-CoV-2 infection. This inhibitory effect has been shown to be due to the oxidizing capacity of the IOL. This oxidizing capacity is exerted towards the virus but does not damage eukaryotic cells either in the in vitro or in vivo skin models. Obtained results indicate that the use of IOL, a hydrophilic liquid mixture saturated with highly reactive oxygen and nitrogen species, is a new powerful, safe, and effective tool for preventing possible future outbreaks of the COVID-19 disease.

## 1. Introduction

The COVID-19 pandemic has been countered with a combination of specific and non-specific preventive strategies [1]. Specific measures encompass the use of anti-viral monoclonal medications and active immune prophylaxis, which primarily involves vaccinations. On the other hand, non-specific prophylaxis revolves around individuals employing personal protective equipment and ensuring environments are properly sanitized. Each of these strategies works in tandem, complementing one another [2]. However, the epidemiological landscape continues to shift due to the emergence of new mutations and variants of the SARS-CoV-2 virus. This evolving nature raises concerns over the potential development of escape mutants, which might not be detected or neutralized by the current immunization methods in place [3].

The stability of the SARS-CoV-2 virus in the environment varies depending on the type of surface and surrounding conditions. Environmental factors like temperature, humidity, and sunlight also play a role in the virus’s stability. Furthermore, another study in “The Lancet Microbe” showed that the virus’s stability decreases with an increase in temperature. This variability highlights the importance of regular cleaning and disinfection, especially on frequently touched surfaces, to curb the transmission of the virus [4].

Throughout the COVID-19 pandemic, households have been key transmission sites. SARS-CoV-2 has been identified on surfaces in homes, especially where prolonged contact occurs with infected individuals. Unlike hospitals, data on household SARS-CoV-2 contamination remain limited [5,6].

Whereas, research has shown widespread SARS-CoV-2 surface contamination in hospitals treating COVID-19 patients. In Singapore, over half of the isolation rooms had at least one contaminated surface, with a significant percentage of rooms showing contamination on frequently touched surfaces during a patient’s initial illness week. A Toronto study found 26% of surface samples in over half of patient rooms tested positive for the virus. Another study detected the virus in hospital air exhausts, indicating that virus-laden droplets might be carried by airflows onto equipment [7].

Some COVID-19 patients seem to contaminate their surroundings more than others. Factors such as higher respiratory viral loads, the severity of hypoxia upon admission, comorbidity scores, and duration since illness onset are linked to greater surface contamination [7].

There are several considerable factors that affect the viability of SARS-CoV-2 in the environment. The persistence of SARS-CoV-2 varies depending on the type of surface. Generally, coronaviruses last longer on nonporous materials than porous ones. Surfaces like surgical masks, which are hydrophobic and synthetic, show longer virus persistence than absorbent materials like cotton [8]. The virus’s stability is also influenced by its surrounding medium. For instance, when proteins like bovine serum albumin or mucus are added, the virus remains more infective, suggesting airway secretions might enhance its longevity and transmission [9,10]. SARS-CoV-2’s stability decreases with increasing temperature. For instance, its half-life is 1.7–2.7 days at 20 °C but only a few hours at 40 °C. At colder temperatures, like 4 °C, it can persist for up to 14 days, but at 56 °C and 70 °C, its survival time reduces to 10 min and 1 min, respectively [11]. Notably, the virus remains stable and more infectious at freezing temperatures (−20 °C) and can last for 60 days on cold-chain food packaging kept below −18 °C [12]. The relationship between the virus’s viability and relative humidity is intricate. Some findings indicate that SARS-CoV-2 decays fastest at 65% humidity and more slowly at both lower (40%) and higher (75%) levels. The rate of virus inactivation increases with temperature and is also influenced by relative humidity, creating a U-shaped relationship [13].

The effectiveness of chemical disinfectants varies based on the virus structure and the environment. Inappropriate choice and incorrect usage can contribute to pathogen spread, raising public health issues. Therefore, thorough studies are essential to ensure the right selection and application of disinfectants [14]. Alcohols, particularly ethanol and isopropanol, destroy microorganisms by dissolving lipid membranes and denaturing proteins, making them effective against a variety of pathogens, including SARS-CoV-2 [15]. Quaternary ammonium compounds (QACs) are frequently used as disinfectants in healthcare, food processing, and households [16]. Recent studies confirm QACs’ efficacy against SARS-CoV-2, making them a dominant disinfectant on the EPA’s List N [17]. Sodium hypochlorite, a key ingredient in household bleach and other commercial products, is recommended by health organizations for disinfecting surfaces, especially in COVID-19-exposed areas. A “strong chlorine solution” is notably effective against SARS-CoV-2 within minutes. Various studies confirm its efficacy against SARS-CoV-2, though results differ based on concentration and contact time. Notably, while hypochlorous acid is cost-effective and generally safe for many applications, including mouthwashes and sanitizers, excessive chlorine can have environmental and health implications [18,19]. Some authors found that disinfectants based on 83% ethanol, 60% propanol/ethanol, sodium dichloroisocyanurate, and 0.5% potassium peroxymonosulfate inactivated SARS-CoV-2 effectively and safely. Although disinfectants based on 0.05–0.4% benzalkonium chloride (BAC), 0.02–0.07% quaternary ammonium compound (1:1), 0.4% BAC/didecyldimethylammonium chloride, 0.28% benzethonium chloride concentrate/2-propanol, and 0.5% hydrogen peroxide inactivated SARS-CoV-2 effectively, they exhibited cytotoxicity [20].

Hydrogen peroxide is a popular disinfectant known for its environmental safety, as it breaks down into water and oxygen. Its non-toxic nature makes it suitable for disinfecting medical equipment, surfaces, and even skin [20]. Additionally, hydrogen peroxide can be vaporized for fumigation, which is particularly useful for decontaminating hard-to-reach areas, like entire rooms in hospitals. This gaseous form enhances its biocidal activity. Its disinfecting action arises from the generation of hydroxyl free radicals that damage microbial lipids, proteins, and DNA, with its small molecular size allowing it to penetrate microbial defenses without inducing lysis [20]. While disinfectants like hydrogen peroxide, sodium hypochlorite, quaternary ammonium compounds, and alcohols are effective against pathogens like COVID-19a, their widespread use in the environment raises concerns. These agents can have deleterious effects on surfaces, often leading to discoloration, corrosion, and degradation, especially for delicate or specialized materials [19,21]. SARS-CoV-2 is highly sensitive to oxidizing agents due to the pronounced electrophilicity of its spike protein. SARS-CoV-2’s contagion levels peak during winter, with reduced air dilution and increased levels of pollutants like sulphur dioxide [21]. In contrast, summer sees a drop in cases due to improved air dilution and prevalent oxidizing pollutants like ozone. Sanitizing confined areas with oxidizing agents could be beneficial [22]. H_2_O_2_ solutions with no additives displayed a scarce virucidal activity (1.1 log10 diminution in 5 min), confirming that a pH-modifying ingredient is necessary to have a H_2_O_2_-based disinfectant active against the novel coronavirus [23]. Indeed, three decontamination methods, UV irradiation, vaporised H_2_O_2_, and dry heat treatment, for inactivating an infectious non-enveloped virus, in line with the FDA policy regarding face masks, have been studied by some authors [24]. The aim of this study is to assess the efficacy and safety of liquid hyperoxygen (IOL), an aqueous solution saturated with hydrophilic oxygen and oxidizing nitrogen species, as an antiviral agent against SARS-µCoV-2 viruses.

## 2. Materials and Methods

### 2.1. Liquid Hyperoxygen (IOL)

IOL is an aqueous electrolytic solution characterized by radical oxidative properties, attributable to the activity of a single chemical species present as solutes, while the solvent consists of pyrogenic bi-distilled water for liquid phase injections.

IOL is an aqueous mixture of reactive oxygen and nitrogen species in which the gas transmitter molecules derived from the dioxygen and nitrogen oxide have sufficient chemical stability, suitable for in vitro studies of cell physiology. The IOL production technique is based on the generation of an electron beam obtained through an inverse sputtering electron device. The result is a gaseous mixture of allotropes of both oxygen and nitrogen in trace amounts, later dissolved in an aqueous phase. The concentration of the NO radical in IOL is 50 µM, the hydroxyl radical (OH●) is 3.5 µM, and the anion superoxide radical (O_2^−^_) is 0.5 µM, as evaluated by spectroscopic and spectrofluorimetric ion chromatographic, and UV spectroscopic and conductimetric analyses [25].

The chemical–physical characteristics of the water used as a solvent to produce IOL solutions express a conductivity between 20 and 30 µS × cm^−1^ and impurities that never exceed 1 µM/L, and a latent heat of vaporization equal to 9.780 kcal mol^−1^ with 373.15 K as the boiling point. The solutes contained in the IOL aqueous solution are composed of highly reactive species of oxygen and nitrogen (RONS). The RONS, in IOL aqueous solutions, are dissolved as gaseous solutes in the aqueous solvent, reaching maximum concentration values close to 0.98 mg/mL, values which depend on both the temperatures and the pressures; however, the RONS concentration values never exceed 1 mg/mL. Most of the solutes in the IOL aqueous solution are derived from molecular oxygen reduction intermediates and, due to their outermost electron orbital, are occupied by a single electron and included in the group of molecules “one-electron oxidants” (OSE). Among the highly reactive species of oxygen, the most represented in the mixture, besides dioxygen (O_2_), is the superoxide anion (O_2^−^_), while the hydroxyl radical (OH), singlet oxygen (O21∑g+), and hydrogen peroxide (H_2_O_2_) are present only in trace amounts. However, it is important to point out that the presence of nitric oxide (NO) and peroxynitrite (ONOO-), even if present in trace amounts, represent a peculiar characteristic of aqueous IOL solutions and some biological activities of the solutions are due to them, primarily antiviral ones. Neither stabilizing agents nor traces of metals are contained in IOL [25].

IOL is produced by Barcoline srl, Pisa, Italy and distributed by Porti Verdi, Pisa, Italy (patent n. PCT/IB2017/056565 srl).

The IOL stock solution (kindly provided by Porti Verdi, Pisa, Italy) was diluted in physiological solution (0.9% NaCl in molecular grade bi-distilled water) 1/50, 1/100, 1/1000, and 1/10,000 *vol*/*vol*. Incubation times between IOL and throat samples containing SARS-CoV-2 were 1, 5, 10, 20, and 30 min.

### 2.2. Evaluation of Liquid Hyperoxygen Efficacy: Challenge Assay with SARS-CoV-2 Virus

Ten throat swab samples containing the viable SARS-CoV-2 virus were collected from COVID-19-affected patients, 5 composed of omicron and 5 delta variants. Informed consent was obtained from all subjects at the San Martino Hospital, Genoa, Italy. The persistence of the virus and the ability of these samples to infect susceptible cells equipped with specific receptors for this virus were then verified. The presence or absence of the virus inside these cells was evaluated by qPCR. In particular, the biological test used was the ‘challenge test’ [21,22]. This test uses kidney cells with high membrane expression of the ACE2 receptor, which specifically binds the spike protein of SARS-CoV-2 and is therefore able to verify not only the presence of the virus but also above all its infective and pathogenetic capacity.

All the experiments were carried out in the BSL3 biosafety laboratory and carried out by personnel equipped with adequate personal protective equipment suitable for the COVID-19 biological risk (Appendix A).

Cells expressing the high-affinity ACE2 receptor for the SARS-CoV-2 spike protein were incubated overnight at 37 °C with the high viral load samples variously treated with IOL. At the end of the incubation, the plates were subjected to heating at 57 °C for 30 min to inactivate the extracellular virus and to detach the cells from the adhesion plane. The cell suspension was then collected by scraping and aspiration with disposable Pasteur pipettes and then centrifuged for 15 min at 3000 g. The harvested cell pellet was resuspended and washed 2 times with phosphate-buffered saline (PBS) and re-centrifuged. The final collected pellet was then resuspended in sterile RNAse-free molecular grade bi-distilled water and frozen. Intracellular RNA extraction was performed by high-performance robotic equipment using the Janus G3 preparatory robot (PerkinElmer, Milan, Italy) and extraction with magnetic beads using the automated Chemagic 360 D equipment (Perkin Elmer). The presence of SARS-CoV-2 viral RNA in the extracted intracellular RNA was verified by RNA-DNA reverse transcription and polymerization chain reaction (PCR) using a high-sensitivity Light Cycler II apparatus (Roche, Basel, Switzerland).

For the analysis of each sample, molecular fluorescent probes were used for the following genes: (a) gene housekeeping house- Ribonuclease P/MRP Subunit P30 [RPP30] used as an internal positive control to verify the presence of RNA and the correct actuation of the PCR reaction; (b) SARS-CoV-2 Opening Reading Frame (Orf) viral gene Orf1ab labeled with Vic fluorescent probe; and (c) SARS-CoV-2 N viral gene labeled with FAM fluorescent probe. The following PCR amplification time/temperature conditions were used: 50 °C × 15 min, 95 °C × 2 min, 45 cycles at 95 °C × 3 s and 60 °C × 30 s.

### 2.3. Evaluation of Liquid Hyperoxygen Efficacy in Oxidizing Biological Fluids

The ability of IOL to exert an oxidizing action on complex biological fluids and matrices was determined and quantified by the FRAS test (Free Radical Analytical System) on human plasma, using FRAS 5 EVOLVO (H&D, Parma, Italy).

### 2.4. Evaluation of Liquid Hyperoxygen Safety

The safety of IOL was verified by the following: (a) in vitro evaluation of the absence of cellular cytotoxicity by comparison with the cytotoxic effects of other chemical disinfectants (benzalkonium chloride) evaluated with phase contrast optical microscopy; and (b) the absence of in vivo cytotoxicity by human skin erythematogenesis test. To use IOL for environmental disinfection, the possibility of using oxidation-sensitive colorimetric indicators sensitive to oxidation was evaluated to determine the achievement of the effective dose of IOL suitable for neutralizing SARS-CoV-2. Each analysis was tested in triplicate in 3 independent experiments.

## 3. Results

### 3.1. Efficacy

The efficacy of IOL in neutralizing the ability of SARS-CoV-2 to infect sensitive cells was determined by comparing the amount of viral RNA present inside the cells after the incubation with samples containing the virus or with the same samples treated with IOL. The viral load was quantified by the PCR positivity cycle. The greater the amount of virus, the fewer amplification cycles were needed to detect it. The raw data obtained by comparing the intracellular viral loads by PCR after incubation with untreated or IOL-treated samples for different times (1–30 min) are shown in Table 1.

The level of viral load present in the sample analyzed is synthetically highlighted with a color code: green, virus not present; orange, virus present with low viral load (positivity ≥ 28th PCR cycle); or red, virus present with high viral load (positivity ≤ 23rd PCR cycle). The analysis did not detect the presence of the virus in the negative controls (C−s). Instead, the presence of high viral loads was tested in all positive controls (C + s). Treatment with IOL significantly decreased the viral load present. A decrease (from red to orange) occurred in the case of incubations for very short periods (1 min). In this experimental condition, however, the viral load was reduced by as much as 10 orders of magnitude in the case of IOL diluted 1/50 (Exp 2 from 18.8 to 28.8) and by eight orders of magnitude in the case of IOL diluted 1/1000 (Exp 3 from 21.3 to 28.9). For longer incubation times (5–30 min) the high viral load present (PCR cycle positivity 18–21) was always completely inactivated. This result was obtained at all the dilutions evaluated (1/50, 1/100, and 1/1000). In the sole case of the 1/10,000 dilution with incubation at 5 min, there was no neutralization of the virus but the viral load decreased by as much as four orders of magnitude (from 21 to 24), despite the high dilution and short incubation time.

In Table 2, the left column indicates the IOL dilution used. Row ‘0’ indicates the positive control not treated with IOL. Each row corresponds to a certain dilution. On the right, each column corresponds to a specific incubation time (0, no incubation, 1–30 min).

The sample not treated with IOL presented a high viral load with positivity at cycle 20.6. Treatment with IOL diluted by 1/10 always completely inactivated the virus, even for very short incubation periods (1 min). At the 1/50 dilution, at least 5 min of incubation was necessary to obtain the neutralization of the virus. At higher dilutions (1/100, 1/1000) neutralization was obtained after 30 min of incubation. At extreme dilutions (1/100,000 after 5 min), a decrease in the viral load of five orders of magnitude (from 20.6 to 25.4) was observed but not complete inactivation of the virus.

All doses evaluated (dilutions 1/10–1/1000) decreased the viral load by eight orders of magnitude. Achievement of the complete neutralization threshold was achieved for the 1/10 dose. The observed dose/response relationship substantiates the specificity of the antiviral effect of IOL on SARS-CoV-2 (Figure 1).

All doses evaluated (1/10–1/1000 dilutions) neutralized SARS-CoV-2 with viral load decreases greater than 10 orders of magnitude. At the extreme dilution of 1/10,000, the 5 min incubation caused a reduction in the viral load of five orders of magnitude, but not the achievement of the threshold of complete viral inactivation. After 30 min of incubation, all doses evaluated (dilutions 1/10–1/1000) neutralized SARS-CoV-2 with decreases in viral load greater than 10 orders of magnitude (Figure 1).

The findings suggest that IOL exhibits a significant ability to inhibit the SARS-CoV-2 virus. This virus’s capacity to infect susceptible cells was swiftly neutralized, even at minimal concentrations (dilution 1/1000) and brief exposure times (5 min). These results are due to the sensitivity of the SARS-CoV-2 virus towards oxidizing agents. Indeed, this virus, due to the distance between the trans-peri capsid spike proteins, exposes large areas of its lipid mantle to peroxidation induced by oxidizing disinfectants. Furthermore, the viral spike protein is highly electrophilic (positive electrical charge) and, therefore, easily denatured by negatively charged oxidizing agents, such as superoxide anion (O_2^−^_) and related reactive oxygen species.

### 3.2. Evaluation of IOL Stability over Time

To check IOL stability over time, the antiviral efficacy of IOL against the SARS-CoV-2 virus was repeated after 6 months of IOL storage at room temperature (indoor storage in a climatized environment from May to October, indoor temperature 22–25 °C). The IOL dilution tested was 1/50 *v*/*v* and the incubation time with samples containing the virus, set up as previously reported, was 30 min. At time 0 (May), the qPCR positivity threshold cycle (Ct) was 20.6 in positive control samples (untreated with IOL) and 34 in positive samples treated with IOL. The same analysis, repeated after 6 months, i.e., in October, provided a qPCR positivity threshold cycle (Ct) at 20.1 in positive control samples (untreated with IOL) and 33.2 in positive samples treated with IOL. The delta Ct between the control positive sample and the IOL treated samples at T0 was 13.4 and at T1 was 13.1, i.e., 6 months later. Accordingly, IOL antiviral activity undergoes only a 2.24% decrease after 6 months of storage at room temperature. These results demonstrate that the IOL mixture is quite stable when stored at room temperature.

### 3.3. Quantification of the Oxidizing Capacity of IOL on Human Serum

Aliquots (500 uL) of human blood serum were collected from peripheral capillary blood by puncturing the fingertip with a lancet from a volunteer subject (58-year-old male). The IOL was then added to the samples, reaching maximum concentration values of RONS close to 0.98 mg/mL (10 uL dilutions 1/10, 1/100, 1/1000, and 1/10,000). The evaluation of the total oxidative load was carried out by means of analysis of the oxidation capacity of the iron and photometric detection at 522 nm of the oxidized iron. The Fras 5 Evolvo system (H&D, Parma, Italy) was used, consisting of a centrifuge, a thermostatic incubator at 37 °C and a photometer. The determined oxidative load is expressed as equivalents of hydrogen peroxide (U Carr). The results obtained are shown in Table 3.

The oxidative load value detected in the untreated sample was equal to 300 U Carr. At the 1/10 and 1/100 dilutions, the oxidation induced by IOL brought the detected values outside the instrumental reading scale (>5000 U Carr). The oxidative load increased in a detectable way from 300 to 4947 U Carr using IOL diluted by 1/1000 and to 443 (therefore, with an increase of 33%) using IOL diluted by 1/10,000. These results indicate that IOL can exert a powerful oxidative action, even in the context of complex biological matrices. This result is relevant since viruses, including SARS-CoV-2, are always contained in complex biological fluids that can neutralize the action of disinfectants.

### 3.4. In Vitro Test

To verify the specificity and the action of IOL against SARS-CoV-2 and to neutralize the pathogen without damaging human tissues, we performed the in vitro test. Eukaryotic cells of the Vero line were incubated with IOL at different dilutions and for comparison with hydrogen peroxide. The cytotoxicity of IOL versus H_2_O_2_ 5.6% *vol*/*vol* at 5, 10, 15, 30, 60, and 120 min was comparatively evaluated (Figure 2).

Cell morphology and appearance were dramatically changed in the H_2_O_2_-treated cells as compared to untreated cells (negative control, C−). After H_2_O_2_ treatment, cells detached from basal adherence and do not reflect light anymore, thus assuming a dark color. These changes reflect cell sufferance, decreased cell viability and cell death. Indeed, especially at 120 min, many dead cells are floating in the medium, as detectable by empty circles. None of these changes was detected in the IOL treated cells at any treatment times, including 120 min.

### 3.5. In Vivo Test

In the in vivo test, the erythematogenic capacity of IOL was evaluated and compared with a disinfectant known for its cytotoxicity, i.e., benzalkonium chloride used at 5% *vol*/*vol* [21,22,23]. IOL at different dilutions (1/10, 1/100, 1/1000, and 1/10,000) and benzalkonium chloride were applied on rounded skin areas located on the palmar face of the forearm delimited by a dermographic pen. Before application, the area was cleaned with a saline solution. The solutions were left to act for 1, 5, 15, 30, and 60 min. The results obtained are shown in Figure 3.

Benzalkonium chloride induced the formation of an erythematous halo over the entire area of skin where it was applied (positive control, C+, bottom right photo). At none of the concentrations used, and even for longer application times, IOL induced the formation of erythema. This result demonstrates the safety of IOL compared to other chemical disinfectants, such as benzalkonium chloride.

### 3.6. Applications of Environmental Sensors to Detect the Achievement of Disinfectant Charge of IOL

To establish whether a concentration of IOL is reached in the confined environment suitable for guaranteeing the successful neutralization of SARS-CoV-2, we evaluated the possibility of developing environmental sensors using the ability of chromogenic substrates to detect IOL-induced oxidation. The first sensor evaluated is based on a chromogenic liquid containing iron which is oxidized by the interaction with IOL changing its color from transparent to pink to red to carmine red depending on the level of oxidation induced. This test (PAT test) is commercially available (H&D Parma, Italy) and is described in detail by Benedetti et al., 2013 and Iannitti et al., 2012) [26,27]

An example of the application of this sensor is shown in Figure 4.

The results obtained indicate that the color of the liquid sensor becomes darker and more intense with increasing concentrations of IOL used with dark red colors at concentrations of 1/10 and 1/100, dark pink at 1/1000, and light pink at 1/10,000.

Although quite sensitive, the liquid sensor presents some problems in the practical application in a confined environment: (a) its positioning does not appear simple due to the possibility of spilling the liquid since the test tube where the liquid is placed must be opened to interact with diffuse IOL in the air; (b) the interaction surface between the liquid sensor and the diffused IOL aerosol coincides with the liquid surface only and is therefore rather small; and (c) the penetration capacity of the IOL aerosol inside the test tube with the cap present, even if not closed, appears at least questionable.

For all these reasons, we thought it interesting to develop a solid rather than a liquid environmental sensor. The solid sensor consists of a colorimetric paper on which a chromogenic mixture sensitive to oxidation is adsorbed. We evaluated the ability of this sensor to color shift proportionally based on the amount of IOL received. The results obtained, compared to what has already been performed with the liquid sensor, are shown in Figure 5.

The colorimetric paper was very sensitive to IOL-induced oxidation. The map would have several advantages for use as an environmental sensor, in particular the following: (a) it can be loaded on adhesive support for easy and immediate application to all environmental surfaces; (b) the interaction surface with the IOL aerosol is maximized and coincides with the entire sensor surface; and (c) there are no sensor management problems or spillage of the liquids contained therein.

## 4. Discussion

In this study, we investigated the antiviral efficacy and safety of IOL, an aqueous solution saturated with ROS and RNS, against the SARS-CoV-2 virus. Utilizing throat swab samples containing both omicron and delta variants of the virus, we exposed them to varying dilutions of IOL and assessed the remaining viral load using qPCR. Our findings revealed that IOL effectively neutralized SARS-CoV-2 in a dose and time-dependent manner, with notable reductions in viral loads even at higher dilutions. Moreover, in safety assessments, IOL demonstrated no cytotoxic effects on Vero eukaryotic cells and induced no erythema when applied on human skin, in contrast to other disinfectants like hydrogen peroxide and benzalkonium chloride. This suggests IOL’s potential as a specific and non-damaging disinfectant against SARS-CoV-2.

Our experimental procedures involved direct incubation of IOL with live SARS-CoV-2 virus obtained from patient throat swabs and subsequent evaluation of viral viability in cells expressing the ACE2 receptor. Our findings suggest that IOL, even at dilutions of 1/1000, was able to neutralize the infectivity of SARS-CoV-2 in a time-dependent manner, with considerable efficacy evident at shorter incubation times for higher concentrations. Furthermore, the development of a colorimetric paper sensor suggests the potential for real-time IOL monitoring in sanitized environments. Collectively, these results highlight the promising potential of IOL as an effective and safe antiviral agent against SARS-CoV-2.

In recent research on the antiviral effects of various agents on SARS-CoV-2, the use of IOL has emerged as a potential breakthrough. Leveraging its unique composition saturated with hydrophilic oxygen and oxidizing nitrogen species, IOL demonstrated a robust capacity to neutralize SARS-CoV-2 in vitro, which was assessed using a ‘challenge test’ with cells expressing the high-affinity ACE2 receptor, pivotal for the virus’s entry. These findings highlighted a dose-dependent and time-sensitive efficacy of IOL, where even at extreme dilutions such as 1/10,000, there was a significant viral load reduction, with complete inactivation evident at lesser dilutions after extended incubation. The study also emphasized IOL’s safety profile, evidenced by its non-cytotoxic nature with Vero cells and the absence of erythematous reactions on human skin, a stark contrast to agents like benzalkonium chloride. However, it is critical to compare these findings with previous research to determine the relative efficacy, specificity, and safety of the IOL compared to other potential antiviral agents.

The ongoing battle against the COVID-19 pandemic has seen numerous preventive strategies employed, both specific and non-specific [24,28]. Studies have outlined the persistence of the virus on diverse surfaces and under different temperature conditions. It has been reported that SARS-CoV-2 remains viable on surfaces, such as plastic and stainless steel, for up to 2–3 days [4]. Our current findings reveal a potent antiviral effect of IOL, an innovative solution that challenges the traditionally accepted duration of SARS-CoV-2 viability on surfaces.

Homes have been identified as significant transmission sites, with previous studies showing 46% of household surfaces testing positive for SARS-CoV-2 a month after symptoms receded [27,29].

In outpatient settings, prior studies have identified contamination on diverse fixtures and equipment [28,30]. The high contamination rate in hospital settings, such as the study showing more than half of isolation rooms having at least one contaminated surface [29,31], emphasizes the need for effective disinfection methods. In our research, when surfaces were treated with IOL, a notable decline in detectable SARS-CoV-2 was observed.

General observations from previous studies indicate that coronaviruses have a longer survival rate on nonporous materials, like surgical masks, than on absorbent ones like cotton [32]. However, the substantial spacing between the trans-pericapsid spike proteins renders large sections of its lipid mantle susceptible to peroxidation by oxidizing agents. Moreover, the prominently electrophilic nature of the viral spike protein facilitates its denaturation by anionic oxidizing entities like superoxide anion (O_2^−^_) and peroxynitrite (ONOO-). Even though the latter appears minimally in the radical aqueous blend, their presence significantly influences the virus’s susceptibility to IOL. Therefore, IOL’s potential as a disinfectant in real-world settings, such as households, hospitals, and clinics, is accentuated, especially when integrated with colorimetric sensors to confirm adequate disinfection levels, ensuring both efficacy and safety [33,34].

While our study demonstrates the antiviral efficacy of IOL against SARS-CoV-2 viruses, it is pertinent to acknowledge certain limitations. Firstly, our research relied on in vitro tests using the Vero cell line, which might not fully replicate the in vivo dynamics and interactions of the virus in human tissue. The study focused on two SARS-CoV-2 variants (omicron and delta) and may not represent the antiviral activity against other variants or future mutations. The use of PCR cycle thresholds as a surrogate measure for viral infectivity, while practical, may not capture the complete infectious potential of residual viral particles. Additionally, the oxidative action of IOL on complex biological fluids was mainly assessed with human plasma, which might not necessarily extrapolate its efficacy to other biological matrices. Lastly, the assessment of IOL’s safety, although comprehensive, was restricted to short-term observations, leaving potential long-term effects unexplored.

Given the demonstrated efficacy of IOL in neutralizing the SARS-CoV-2 virus, even at significant dilutions, coupled with its noted safety profile in both in vitro and in vivo models, it becomes a promising candidate for environmental disinfection in confined spaces, hospitals, clinics, and households to limit the spread of the virus. Furthermore, the development of solid colorimetric sensors to detect the presence of IOL ensures effective disinfection while negating any cytotoxic effects. It is noteworthy that liquid oxygen derivatives have been already proposed, because of their safety and efficacy, for medical treatments [35,36]. In principle, due to its safety and efficacy, an IOL aerosol could be used to disinfect air and surfaces in indoor spaces. However, the focus of this work was mainly on in vitro tests and, to a minor extent, on in vivo trials.

Peroxynitrite species are present in traces in IOL. The direct toxicity of nitric oxide is modest but might be enhanced by reacting with superoxide to form peroxynitrite [36,37]. The absence of cytopathic effects reported in our study is likely related to the minimal dose of nitric oxide present in IOL, allowing anti-viral effects but not cytopathic effects. Indeed, at variance with the virus, eukaryotic cells are well equipped with phase II metabolic reactions and intracellular antioxidants able to neutralize peroxynitrite. Peroxynitrite detoxification is furthermore enhanced in vivo where it mainly occurs through isomerization by oxyhemoglobin in red blood cells [38].

## 5. Conclusions

The results obtained from the herein-reported experimental study indicate that the use of IOL is a new, safe and effective tool for combating the COVID-19 pandemic. IOL has been shown to be particularly effective, both in the asepsis of mucous membranes positive for COVID-19 and for topical cutaneous prevention treatment. The virus’s capacity to infect susceptible cells is neutralized even at minimal concentrations (dilution 1/1000) and brief exposure times (5 min). This action is carried out selectively without inducing cytopathic effects on healthy cells or erythematogenic effects on the skin.

Furthermore, the use of IOL has been proven to be useful in the disinfection of indoor spaces intended for public use, such as shops, restaurants, shopping centers, etc., as well as indoor spaces used by fragile subjects such as protected residences and medical surgeries.

## Figures and Tables

**Figure 1 pathogens-13-00244-f001:**
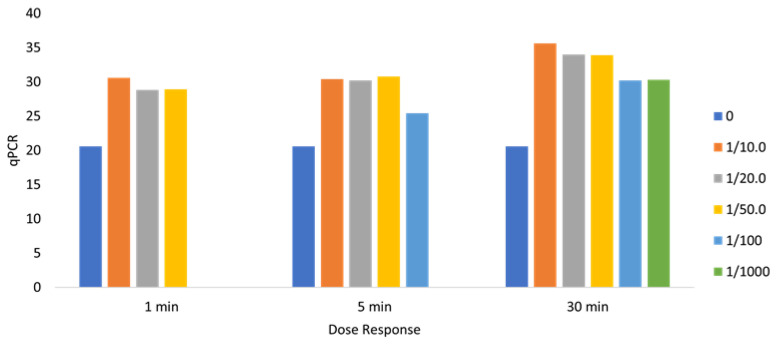
The effect of different IOL dilutions on SARS-CoV-2 (PCR positivity cycle, vertical axis) after incubation for three different time points (horizontal axis). Legends on the right side of the chart represent the IOL dose dilutions. qPCR expression: >30 represents “SARS-CoV-2 not detected, less than >27 represents “SARS-CoV-2 barely detected”, and <27 represents “SARS-CoV-2 detected”.

**Figure 2 pathogens-13-00244-f002:**
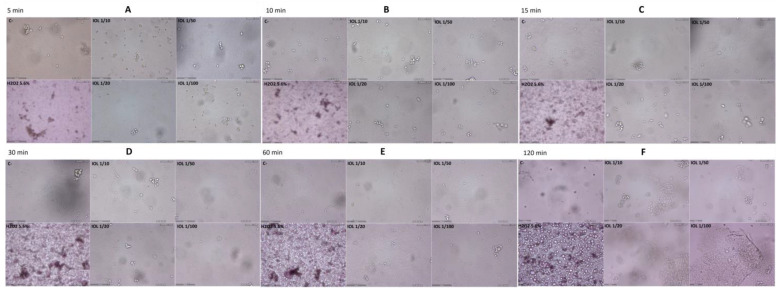
Comparative cytotoxicity of IOL and H_2_O_2_ (5.6% *vol*/*vol*) at time intervals of (**A**) 5 min; (**B**) 10 min; (**C**) 15 min; (**D**) 30 min; (**E**) 60 min; (**F**) 120 min. Vero cells were grown at semiconfluence (control, C, upper left panel in each box) and then exposed to hydrogen peroxide 5.6% *v*/*v* (H_2_O_2_, lower left panel in each box) or IOL at various concentrations *v*/*v* (1/10, 1/20, 1/50, 1/100, four right panels in each box).

**Figure 3 pathogens-13-00244-f003:**
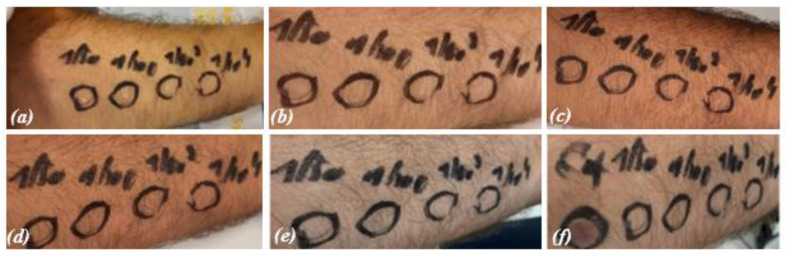
Erythematogenic responses on the forearm after applying IOL dilutions from 1 to 60 min; (**a**) 1 min, (**b**) 5 min, (**c**) 15 min, (**d**) 30 min, (**e**) 60 min, (**f**) C+-positive control (benzalkonium chloride 5 min). A clear erythematogenic response was observed for benzalkonium chloride (C+) left circle in panel (**f**) while no erythomatogenic response was detected for IOL applied at different dilutions (1/50, 1/100, 1/1000, and 1/10,000, circles from left to right in panels **a**–**f**).

**Figure 4 pathogens-13-00244-f004:**
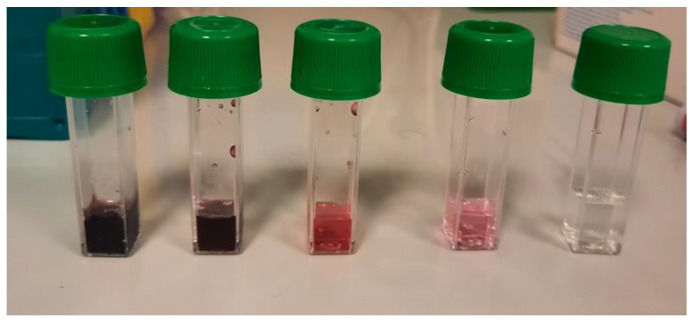
The progressive color transformation of a chromogenic liquid sensor containing iron upon interaction with IOL, transitioning from transparent to varying shades of pink and red based on the oxidation level.

**Figure 5 pathogens-13-00244-f005:**
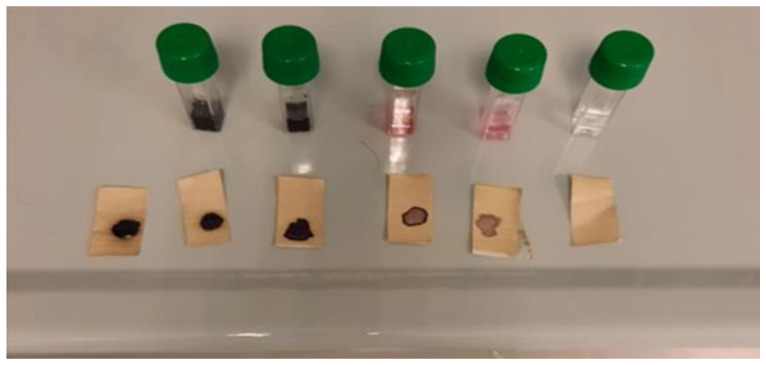
The ability of the solid sensor to shift color proportionally based on the amount of IOL received.

**Table 1 pathogens-13-00244-t001:** Representation of intracellular viral load by PCR with IOL-treated and untreated samples at different time points.

Exp 1	HK Amplicon	Orf1ab Amplicon	N Amplicon	Orf1/N Mean	
C−	28.9	30	30	30.0	**SARS-CoV2** **Not detected**
C+	34.7	22.5	19.5	21.0	**SARS-CoV2** **detected**
IOL 1/10 × 1 min	30.5	30.9	30.4	30.6	**SARS-CoV2** **Not detected**
IOL 1/10 × 5 min	30.7	30.6	30.2	30.4	**SARS-CoV2** **Not detected**
IOL 1/10 × 30 min	30.6	30.0	30.6	30.3	**SARS-CoV2** **Not detected**
IOL 1/10 × 5 min	30.8	30.1	30.3	30.2	**SARS-CoV2** **Not detected**
IOL 1/10 × 30 min	30.8	30.1	30.3	30.2	**SARS-CoV2** **Not detected**
**Exp 2**					
C−	32.9	30	30	30	**SARS-CoV2** **Not detected**
C+	27.1	20.2	17.3	18.8	**SARS-CoV2** **Not detected**
IOL 1/50 × 1 min	32.3	28.6	29.1	28.8	**SARS-CoV2** **Not detected**
IOL 1/50 × 5 min	29.5	30.6	30.9	30.8	**SARS-CoV2** **Not detected**
IOL 1/50 × 30 min	30.4	32.0	32.0	32.0	**SARS-CoV2** **Not detected**
IOL 1/100 × 30 min	31.7	31.8	32.1	32.0	**SARS-CoV2** **Not detected**
**Exp 3**					
C−	29.6	30	30	30	**SARS-CoV2** **Not detected**
C+	29	20.7	21.8	21.3	**SARS-CoV2** **Not detected**
IOL 1/10,000 × 1 min	26.5	32	25.7	28.9	**SARS-CoV2** **Not detected**
IOL 1/10,000 × 5 min	43.9	25.8	25.1	25.4	**SARS-CoV2** **Not detected**
**Exp 4**					
C−	30.2	30	30	30	**SARS-CoV2** **Not detected**
C+	24.1	20.4	21.1	20.8	**SARS-CoV2** **Not detected**
C+	24	20.9	21.2	21.1	**SARS-CoV2** **Not detected**
IOL 1/50 × 30 min	25.6	30	30	30	**SARS-CoV2** **Not detected**
IOL 1/50 × 30 min	25.5	37.9	39.5	38.7	**SARS-CoV2** **Not detected**
IOL 1/100 × 30 min	25.4	37.6	36.4	37	**SARS-CoV2** **Not detected**
IOL 1/100 × 30 min	25.6	35.6	30	32.8	**SARS-CoV2** **Not detected**
IOL 1/1000 × 30 min	24.4	34.8	34.3	34.5	**SARS-CoV2** **Not detected**
IOL 1/1000 × 30 min	25	36.9	36.2	36.6	**SARS-CoV2** **Not detected**

**Table 2 pathogens-13-00244-t002:** The averages of the results obtained in the experiments aimed at establishing the concentration of IOL necessary to neutralize SARS-CoV-2.

	0 min	1 min	5 min	30 min
0	20.6			
1/10	20.6	30.6	30.4	30.3
1/20	20.6		30.2	30.2
1/50	20.6	28.8	30.8	34
1/100	20.6			33.9
1/1000	20.6	28.9		35.6
1/10,000	20.6		25.4	

**Table 3 pathogens-13-00244-t003:** Oxidative load in 10 dilutions (1/10, 1/100, 1/1000, and 1/10,000).

Dilutions	Oxidative Load
µL 1/10	>8000 U Carr
µL 1/100	>8000 U Carr
µL 1/1000	4947 U Carr
µL 1/10,000	443 U Carr

U Carr = 0.08 mg/dL H_2_O_2_ water solution.

## Data Availability

The datasets used and/or analyzed during the present study are available from the corresponding author upon reasonable request.

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
