# Peer review of "Indoor Inactivation of SARS-CoV-2 Virus by Liquid Hyperoxygen"

_pathogens, 2024, doi:10.3390/pathogens13030244_

Round 1

Reviewer 1 Report

Comments and Suggestions for Authors

H2O2 is a known inactivation chemical to enveloped virus like SARS, so please examine the inactivation test with non-enveloped virus.

Your paper sounds good.

Comments on the Quality of English Language

I have no comments.

Author Response

Reviewer 1:

Comment 1: H2O2 is a known inactivation chemical to enveloped virus like SARS, so please examine the inactivation test with non-enveloped virus.

Your paper sounds good.

Answer 1: We thank the Reviewer for the positive comments and the suggested annotations. We have added a reference that reported the H2O2 and the inactivation test with non-enveloped virus.

Reviewer 2 Report

Comments and Suggestions for Authors

The manuscript deals with the development and optimization of use of an aqueous formulation (liquid hyperoxygen 16 (IOL) to be used as an antiviral agent against SARS-CoV-2 virus.

The text is rich in experimental data and covers several aspects of the potential use of the IOL medium as a virucidal solution: efficacy tests on the coronavirus, evaluation tests in human serum, in vivo and in vitro tests as well as additional tests on a colorimetric method to detect the presence of IOL in the environment.

The obtained results are promising and show that IOL is an efficient disinfecting agent against SARS-CoV-2, at very low concentrations and in very short contact times too.

As far as the results are concerned, the manuscript might deserve publication. However, many unclear points are present, that need to be clearly addressed before accepting it. In particular:

1) The most relevant point of weakness is the information about IOL medium. Its description in section 2.1 is quite vague. What is LOH, reported in the title? Is it different from IOL? The physico-chemical properties of the aqueous solvent are carefully described, whereas the nature of the active species is totally unclear and doubtful (“most of the solutes”, “some highly RNS”, “traces”, etc.). Other sentences are totally confused (what is “the foreign Anglo-Saxon language”?). The genesis of this IOL medium is unclear as well. Is it a commercial solution? IOL-RONS is a trademark, apparently. What is the role of “Porti Verdi, Pisa”? Is it a manufacturer or a location where the medium is obtained? Is the stock solution a unique batch? Is it a non-commercial sample? How reproducible is such preparation? If no clear indications are given to the reader about the physical and chemical characteristics of this formulation, it is often hard to assess the reliability of the statements in the following sections of the manuscript. For instance, the Authors affirm peroxynitrite species are present in traces (line 145). Nevertheless, Ref. 21, that they cite themselves, says that “the direct toxicity of nitric oxide is modest but is greatly enhanced by reacting with superoxide to form peroxynitrite”. So, the absence of detrimental effects that is noted in the (partial) tests in sections 3.3 and 3.4 could be a misleading statement, hiding other long terms effects noted by Beckmann in Ref. 21, when NO and ONOO- are present at the same time. Moreover, the vague indications about the composition of the medium do not allow other readers to reproduce and test the effectiveness of the system. Uncomplete information may occur when patented formulations are dealt with. But here no patents are mentioned in the Reference section nor in the main text.

2) The word “environmental” in the title is misleading, as it suggests the use of the disinfectant for environmental sustainability reasons. Actually, it is designed to be tested in confined spaces.

3) Line 87. The sentence “As cationic detergents” has no verb and is meaningless. Please, clarify.

4) Line 110. Oxidising agents typically are not electron donors. They are, rather, electron withdrawing species. They do not necessarily have a negative charge either (see, for instance, cerium(IV) species).

5) There is a limited overview of the previous state of the art on this topic. The Reference section reports only 1 paper published in 2023, but it is not relevant to theme of disinfectants. Covering the area of coronavirus inactivation, novelties in the literature has been frequent in the last few years. Some previous insightful works on the use of hydrogen peroxide (that apparently is a component if IOL) have been overlooked: DOI: 10.1016/j.jhin.2022.09.011; 10.1021/acs.chas.0c00095; 10.1016/j.jhin.2020.09.001, or the preprint work: 10.21203/rs.3.rs-1944446/v1. In particular, the work by the Italian research team describes the effective use of diluted hydrogen peroxide solutions or peroxide salts, with no need of further additives. These previous reports should be taken into account and their performance compared to the one observed with the present text. The statements at lines 115-118 should be revised accordingly.

6) Line 140. One-electron oxidants are quite unstable in aqueous solution, especially in non-degassed solution (leading to termination of the free-radical cascade). Were stabilising agents added to the formulation? Tiny traces of metals lead to rapid abatement of the oxidising capability as well. Do the Authors check for such a presence?

7) Line 147. Was the total content of oxidants titrated in the stock solution? A simple iodometric titration can fit the case. Is the oxidant capability stable over time (in terms of hours or days)?

8) Line 153. Omicron and delta variants of the virus were studied. However, on line 419 it is stated that “wild” SARS-CoV-2 was studied. This terms generally applies to the original wild strain derived directly from the Wu Han outbreak. This is puzzling. Please, clarify.

9) Fig. 1 can be moved to Supplementary Information material.

10) Table 1. OPL acronym is not explained.

11) Figures 2 to 4 are bulky. The use of a more compact histogram is advisable.

12) Table 3. I guess “ul” is microlitre, with Greek letter “mu”. Then, rather than “error” it is preferable to express the value as “more than…” or “above response limit”.

13) Fig. 5 is not self-explanatory. A broader description the observed results is needed for the readers who are not familiar with this kind of in vitro tests.

14) Line 289. What is “epithelial benzalkonium chloride”?

15) Figure 6. Captions on the dermal spots ate not easy to read. In addition, what is the contact time for the positive test?

16) Section 3.5. As for point 1), with no indications about the chemical nature and oxidation state of the iron species, the information is meaningless. The IOL oxidant for instance can degrade the ligand to iron of a possible organic-iron complex and the colour change is then to be attributed to degradation of the complex itself and it is not linearly proportional to the concentration of the oxidising species. More details about the nature of the solution are needed. Otherwise, this section can be removed from the text.

17) Line 335. “The colorimetric paper was also very sensitive to IOL-induced oxidation”. Why also? are there other reactants or contaminants which lead to a colour change for this solution?

18) Line 341. “RONS, approximately 0.98 mg/ml”. How was this concentration measured? By which technique?

19) Lines 346-352. This paragraph sounds like a pre-conclusion and it is redundant is this form here.

20) Lines 349-350. The concepts of coulombic attraction (positive-negative electrostatic charges) seem to be mixed with redox interactions between oxidant and reductants species. This is a misleading oversimplification.

21) Lines 386 to 418 are sometimes a repetition of concepts that should be located in the Introduction section. Several sentences should be merged with the indications in the introduction.

22) Lines 443-445. It is affirmed that “the use of IOL has proved to be useful in the disinfection of confined spaces”. However, the main focus of the work was mainly on in vitro tests and, at a minor extent, on in vivo trials. No experimental tests have been carried out in this work. How IOL is useful in the disinfection of confined spaces? as aerosol? as liquid sprayed form? the present work has shown the potential of the solution ingredients, but it did not show how practical can be its use, neither the technological issues linked to its application in confined spaces. If further data are available, please, provide.

After all of these points are carefully addressed, the manuscript could be newly evaluated for acceptance.

Comments on the Quality of English Language

English language is fine. Some sentences only should be rephrased or revised.

Author Response

Dear Editor,

We would like to thank you for considering the manuscript entitled “Environmental Inactivation of SARS-CoV-2 Virus by Liquid Hyperoxygen” by Izzotti A. et al., and for sharing the Reviewers’ comments that certainly helped in improving the quality of the manuscript (ID: 2844626). We appreciated the Reviewers’ comments, and we revised the manuscript accordingly. Please find enclosed to the submission of the revised version of the manuscript the point-by point reply to the Reviewers’ comments.

We hope that this revised version of our MS will be now suitable for publication in the Pathogens.

Accordingly, we prepared a revised version of the manuscript acknowledging Referees’ and Editor’s comments as below specified:

Reviewer 2.

Comment 1a. 1) The most relevant point of weakness is the information about IOL medium. Its description in section 2.1 is quite vague. What is LOH, reported in the title? Is it different from IOL? The physico-chemical properties of the aqueous solvent are carefully described, whereas the nature of the active species is totally unclear and doubtful (“most of the solutes”, “some highly RNS”, “traces”, etc.). Other sentences are totally confused (what is “the foreign Anglo-Saxon language”?). The genesis of this IOL medium is unclear as well. Is it a commercial solution? IOL-RONS is a trademark, apparently. What is the role of “Porti Verdi, Pisa”? Is it a manufacturer or a location where the medium is obtained? Is the stock solution a unique batch? Is it a non-commercial sample? How reproducible is such preparation?

Answer 1a. Typewriting mistakes (LOH, foreign Anglo-Saxon language, etc. ) have been corrected. All details dealing IOL (manufacturer, distributor, record trademark, patent number) are now reported in the text (Matherials and Methods, 2.1. Liquid hyperoxygen).

Comment 1b.

the Authors affirm peroxynitrite species are present in traces (line 145). Nevertheless, Ref. 21, that they cite themselves, says that “the direct toxicity of nitric oxide is modest but is greatly enhanced by reacting with superoxide to form peroxynitrite”. So, the absence of detrimental effects that is noted in the (partial) tests in sections 3.3 and 3.4 could be a misleading statement, hiding other long terms effects noted by Beckmann in Ref. 21, when NO and ONOO- are present at the same time. Moreover, the vague indications about the composition of the medium do not allow other readers to reproduce and test the effectiveness of the system. Uncomplete information may occur when patented formulations are dealt with. But here no patents are mentioned in the Reference section nor in the main text.

Answer 1b. IOL Patent number is now reported (Matherials and Methods, 2.1. Liquid hyperoxygen) . The matter of possible long term peroxynitrite toxicity is now reported and commented in Discussion (lines).  473-480). A neew reference (32) to better clarify this point has been also added.

Comment 2. The word “environmental” in the title is misleading, as it suggests the use of the disinfectant for environmental sustainability reasons. Actually, it is designed to be tested in confined spaces.

Answer 2. We thank the Reviewer for the annotations. We have changed the title. The new is: “Indoor Inactivation of SARS-CoV-2 Virus by Liquid Hyperoxygen”.

Comment 3. Line 87. The sentence “As cationic detergents” has no verb and is meaningless. Please, clarify.

Answer 3. The sentence has been deleted.

Comment 4. Line 110. Oxidising agents typically are not electron donors. They are, rather, electron withdrawing species. They do not necessarily have a negative charge either (see, for instance, cerium(IV) species).

Answer 4. We thank the Reviewer for the comments and we deleted the sentence.

Comment 5. There is a limited overview of the previous state of the art on this topic. The Reference section reports only 1 paper published in 2023, but it is not relevant to theme of disinfectants. Covering the area of coronavirus inactivation, novelties in the literature has been frequent in the last few years. Some previous insightful works on the use of hydrogen peroxide (that apparently is a component if IOL) have been overlooked: DOI: 10.1016/j.jhin.2022.09.011; 10.1021/acs.chas.0c00095; 10.1016/j.jhin.2020.09.001, or the preprint work: 10.21203/rs.3.rs-1944446/v1. In particular, the work by the Italian research team describes the effective use of diluted hydrogen peroxide solutions or peroxide salts, with no need of further additives. These previous reports should be taken into account and their performance compared to the one observed with the present text. The statements at lines 115-118 should be revised accordingly.

Answer 5. A new paragraph has been added in Introduction/Discussion reporting and discussing published data dealing the use of oxidants as disinfectants against Sars_CoV-2 virus.

Comment 6. Line 140. One-electron oxidants are quite unstable in aqueous solution, especially in non-degassed solution (leading to termination of the free-radical cascade). Were stabilising agents added to the formulation? Tiny traces of metals lead to rapid abatement of the oxidising capability as well. Do the Authors check for such a presence?

Answer 6. The fact that neither stabilising agents or traces of metals is contained into IOL is now reported (Materials and Methods, 2.1 Liquid hypperoxygen, line 156 ).

Comment 7. Is the oxidant capability stable over time (in terms of hours or days)?

Answer 7. It is now reported that ‘IOL is quite stable at room temperature. Oxidizing ability of IOL was tested before and after 6 months of storage at room temperature without observing any significant decrease (line 162-164)’.

Comment 8. Line 153. Omicron and delta variants of the virus were studied. However, on line 419 it is stated that “wild” SARS-CoV-2 was studied. This terms generally applies to the original wild strain derived directly from the Wu Han outbreak. This is puzzling. Please, clarify.

Answer 8. We thank the Reviewer for the comments. The world ‘wild’ have been deleted to avoid any confusion.

Comment 9. Fig. 1 can be moved to Supplementary Information material.

Answer 9. We thank the Reviewer for the suggestion. The Figure has been moved to the supplementary material.

Comment 10. Table 1. OPL acronym is not explained.

Answer 10. OPL has been changed into IOL. Thank you.

Comment 11. Figures 2 to 4 are bulky. The use of a more compact histogram is advisable.

Answer 11. As requested, Figures 2 to 4 has been changed from line pots into histograms and compacted.

Comment 12. Table 3. I guess “ul” is microlitre, with Greek letter “mu”. Then, rather than “error” it is preferable to express the value as “more than…” or “above response limit”.

Answer 12. The symbol is µl. 

Comment 13. Fig. 5 is not self-explanatory. A broader description the observed results is needed for the readers who are not familiar with this kind of in vitro tests.

Answer 13. Fig. 5 legend has been fully re-written giving more explanations. A paragraph has been added describing the results reported in Figure 5.

Comment 14. Line 289. What is “epithelial benzalkonium chloride”?

Answer 14. This erroneous statement has been deleted. Thank you.

Comment 15. Figure 6. Captions on the dermal spots ate not easy to read. In addition, what is the contact time for the positive test?

Answer 15. Caption has been re-written adding more information and explanations including the contact time for the positive test.

Comment 16. Section 3.5. As for point 1), with no indications about the chemical nature and oxidation state of the iron species, the information is meaningless. The IOL oxidant for instance can degrade the ligand to iron of a possible organic-iron complex and the colour change is then to be attributed to degradation of the complex itself and it is not linearly proportional to the concentration of the oxidising species. More details about the nature of the solution are needed. Otherwise, this section can be removed from the text.

Answer 16. Dealing the fact that a more detailed description of IOL composition has been added (see Answer to Comment 1 and new added paragraph in Material and Methods) and no presence of iron species exists, we prefer to leave this paragraph.

Comment 17. Line 335. “The colorimetric paper was also very sensitive to IOL-induced oxidation”. Why also? are there other reactants or contaminants which lead to a colour change for this solution?

Answer 17. We thank the Reviewer for the annotations. We have corrected the sentence.

Comment 18. Line 341. “RONS, approximately 0.98 mg/ml”. How was this concentration measured? By which technique?

Answer 18. The sentence has been rephrased deleting misleading reference to RONS.

Comment 19. Lines 346-352. This paragraph sounds like a pre-conclusion and it is redundant is this form here.

Answer 19. The paragraph has been deleted.

Comment 20. Lines 349-350. The concepts of coulombic attraction (positive-negative electrostatic charges) seem to be mixed with redox interactions between oxidant and reductants species. This is a misleading oversimplification.

Answer 20. The sentence has been deleted.

Comment 21. Lines 386 to 418 are sometimes a repetition of concepts that should be located in the Introduction section. Several sentences should be merged with the indications in the introduction.

Answer 21. The sentence has been moved to Introduction.

Comment 22. Lines 443-445. It is affirmed that “the use of IOL has proved to be useful in the disinfection of confined spaces”. However, the main focus of the work was mainly on in vitro tests and, at a minor extent, on in vivo trials. No experimental tests have been carried out in this work. How IOL is useful in the disinfection of confined spaces? as aerosol? as liquid sprayed form? the present work has shown the potential of the solution ingredients, but it did not show how practical can be its use, neither the technological issues linked to its application in confined spaces. If further data are available, please, provide.

Answer 22. It is now clearly reported that ‘In principle, due to its safety and efficacy, IOL aerosol could be used to disinfect air and surfaces in indoor spaces. However, the main focus of this work was mainly on in vitro tests and, at a minor extent, on in vivo tests’.

Round 2

Reviewer 2 Report

Comments and Suggestions for Authors

The Authors in their rebuttal have only partially replied to Reviewer 2’s points of concern. Most of unclear sentences have been corrected, removed or improved. However, the main point of weakness of the entire manuscript is still pending. In detail:

Comment 1a) The vague description of the composition is still present and has not been substantially changed. Species such as hydroxyl radical or singlet oxygen are not stable species (their half-time is in the order of fraction of seconds) and cannot be considered as ingredients of the medium. They are typically generated in situ when an oxidising precursor reacts with a reducing substrate. Dioxygen (O2) is the common dissolved atmospheric triplet dioxygen that is not reactive to organic substrates and is not an active ingredient. Other statements about the presence of “traces” (lines 160 and 161) are not suitable for a scientific publication. Reviewer’s questions about “Is the stock solution a unique batch? How reproducible is such preparation?” have been left pending.

Comment 1b) The reply about the presence and concentration of peroxynitrite species (a potential carcinogenic species, by the way) is unsatisfactory. How much is a “trace”, taking into account that minor amount can lead to detrimental effects too? The comments at lines 473-480 (unchanged from the previous version) do not cover these issues either. This is a crucial point if this formulation is to be used in healthcare facilities.

Comment 5) The sentence at lines 126-129 is with no verb and is meaningless. It is not clear, therefore, which is the actual advantage of the proposed approach with respect to the current state of the art.

Comment 7) The stability of IOL over time is vague. What does “quite stable” mean? The sentence at lines 170-172 refer to “unpublished data”, but no numerical values are given. How was the oxidising capability measured? Was it titrated? This point of weakness is linked to Comment 1a too.

Comment 12) In table 3, the unit is microlitre, but the symbol “ul” was not changed. The word “error”, that is puzzling as it suggests a statistical deviation or an overflow response from the instrument is still present.

Comment 16) Authors’ reply is puzzling. They state that “no presence of iron species exists”. However, in the text it is affirmed that “The first sensor evaluated is based on a chromogenic liquid containing iron” (line 363) and “chromogenic liquid sensor containing iron” (Fig. 4). The two sentences are in contradiction. Are iron complexes present in the chromogenic detection solution or are they not? Is so, as mentioned in the original Comment 16, with no indications about the chemical nature and oxidation state of the iron species (of the chromogenic solution), the information is meaningless. The method should be reproducible in a scientific paper.

Comment 18) The final paragraphs of section 3.5 was the only section shedding some light into the composition and concentration of oxidising species. It is now removed. So, did the Authors check for the actual amount of oxidising species in IOL?

All of these points still need a careful and focused revision.

Author Response

Comment 1a) The vague description of the composition is still present and has not been substantially changed. Species such as hydroxyl radical or singlet oxygen are not stable species (their half-time is in the order of fraction of seconds) and cannot be considered as ingredients of the medium. They are typically generated in situ when an oxidising precursor reacts with a reducing substrate. Dioxygen (O2) is the common dissolved atmospheric triplet dioxygen that is not reactive to organic substrates and is not an active ingredient. Other statements about the presence of “traces” (lines 160 and 161) are not suitable for a scientific publication. Reviewer’s questions about “Is the stock solution a unique batch? How reproducible is such preparation?” have been left pending.

Comment 1b) The reply about the presence and concentration of peroxynitrite species (a potential carcinogenic species, by the way) is unsatisfactory. How much is a “trace”, taking into account that minor amount can lead to detrimental effects too? The comments at lines 473-480 (unchanged from the previous version) do not cover these issues either. This is a crucial point if this formulation is to be used in healthcare facilities.

Answer 1.

We thank the Reviewer for the annotation. A detailed description of IOL composition have been added (Methods, lines 138-145). Reactive oxygen and nitrogen species content is now reported in nM and uM concentrations, respectively. A reference providing all details of IOL composition and chemical analyses has been added (new added ref. 25 Barco et al., AIP advances, 2024, DOI: 10.1063/50075895).

Comment 5) The sentence at lines 126-129 is with no verb and is meaningless. It is not clear, therefore, which is the actual advantage of the proposed approach with respect to the current state of the art.

Answer 5) We thank the Reviewer. The sentence has been corrected.

Comment 7) The stability of IOL over time is vague. What does “quite stable” mean? The sentence at lines 170-172 refer to “unpublished data”, but no numerical values are given. How was the oxidising capability measured? Was it titrated? This point of weakness is linked to Comment 1a too.

Answer 7) The data dealing the analysis of IOL stability over time are now reported in a newly added paragraph (Results lines 309-319) now specifying that IOL antiviral activity undergo only a 2.24% decrease after 6 months of storage at room temperature.

Comment 12) In table 3, the unit is microlitre, but the symbol “ul” was not changed. The word “error”, that is puzzling as it suggests a statistical deviation or an overflow response from the instrument is still present.

Answer 12) In Table 3, the word ‘error’ was related to the fact that results obtained are above the analysis range of the instrument. For the sake of clarity, the word ‘error’ has now been changed into >8000 U Carr.

Comment 16) Authors’ reply is puzzling. They state that “no presence of iron species exists”. However, in the text it is affirmed that “The first sensor evaluated is based on a chromogenic liquid containing iron” (line 363) and “chromogenic liquid sensor containing iron” (Fig. 4). The two sentences are in contradiction. Are iron complexes present in the chromogenic detection solution or are they not? Is so, as mentioned in the original Comment 16, with no indications about the chemical nature and oxidation state of the iron species (of the chromogenic solution), the information is meaningless. The method should be reproducible in a scientific paper.

Answer 16) The statement that ‘no presence of iron species exists” refers to IOL composition and not to the chromogenic liquid sensor used. Indeed, this senso contains iron whose oxidation change absorbance at 522 nm. This is now specified. Two new references have been added to explain this test .

Comment 18) The final paragraph of section 3.5 was the only section shedding some light into the composition and concentration of oxidising species. It is now removed. So, did the Authors check for the actual amount of oxidising species in IOL?

Answer 18. Please note that this paragraph, according to other reviewers’ request, has not been deleted but moved to paragraph 3.1 lines 300-308.

Round 3

Reviewer 2 Report

Comments and Suggestions for Authors

The Authors have now adequately addressed all issues previously raised. Lastly, I would suggest to check the notation of units and formulae (mM instead of uM or superscripts/subscripts for radical species, especially in Section 2.1). After this, the text might be positively considered for publication.